# Safety and immunogenicity of a subtype C ALVAC-HIV (vCP2438) vaccine prime plus bivalent subtype C gp120 vaccine boost adjuvanted with MF59 or alum in healthy adults without HIV (HVTN 107): A phase 1/2a randomized trial

Zoe Moodie[1]*, Erica Andersen-Nissen[1,2], Nicole Grunenberg[1¤a], One B. Dintwe[1,2], Faatima Laher Omar[2], Jia J. Kee[1], Linda-Gail Bekker[3], Fatima Laher[4], Nivashnee Naicker[5], Ilesh Jani[6], Nyaradzo M. Mgodi[7], Portia Hunidzarira[7], Modulakgota Sebe[8], Maurine D. Miner[1], Laura Polakowski[9], Shelly Ramirez[1], Michelle Nebergall[1], Simbarashe Takuva[1¤b], Lerato Sikhosana[10¤c], Jack Heptinstall[11], Kelly E. Seaton[11], Stephen De Rosa[1], Carlos A. Diazgranados[12¤d], Marguerite Koutsoukos[13], Olivier Van Der Meeren[14], Susan W. Barnett[15], Niranjan Kanesa-thasan[16¤e], James G. Kublin[1], Georgia D. Tomaras[11], M. Juliana McElrath[1], Lawrence Corey[1], Kathryn Mngadi[8☯], Paul Goepfert[17☯], on behalf of the HVTN 107 Protocol Team[¶]

1 Vaccine and Infectious Disease Division, Fred Hutchinson Cancer Center, Seattle, Washington, United States of America, 2 Cape Town HVTN Immunology Laboratory, Hutchinson Centre Research Institute of South Africa, Cape Town, South Africa, 3 Desmond Tutu HIV Centre, University of Cape Town, Cape Town, South Africa, 4 Perinatal HIV Research Unit, Faculty of Health Sciences, University of the Witwatersrand, Johannesburg, South Africa, 5 Centre for the AIDS Programme of Research in South Africa, University of KwaZulu-Natal, Durban, South Africa; and Department of Public Health Medicine, School of Nursing and Public Health, University of KwaZulu-Natal, Durban, South Africa, 6 Instituto Nacional de Saude, Maputo, Mozambique, 7 Clinical Trials Research Centre, University of Zimbabwe College of Health Sciences, Harare, Zimbabwe, 8 Aurum Institute, Tembisa, Gauteng, South Africa, 9 DAIDS, NIAID, NIH, Bethesda, Maryland, United States of America, 10 Hutchinson Centre Research Institute of South Africa, Johannesburg, South Africa, 11 Department of Surgery, Duke University, Durham, North Carolina, United States of America, 12 Formerly Sanofi-Pasteur, Swiftwater, Pennsylvania, Pennsylvania, United States of America, 13 GSK, Wavre, Belgium, 14 Previously GSK, Rixensart, Belgium, 15 Bill & Melinda Gates Foundation, Seattle, Washington, United States of America, 16 Formerly GSK, Rockville, Maryland, United States of America, 17 Department of Medicine, University of Alabama at Birmingham, Birmingham, Alabama, United States of America

☯ These authors contributed equally to this work.
¤a Current address: PATH, Seattle, Washington, United States of America
¤b Current address: School of Health Systems and Public Health, Faculty of Health Sciences, University of Pretoria, Pretoria, South Africa
¤c Current address: National Health Laboratory Service, Johannesburg, South Africa
¤d Current address: Bill & Melinda Gates Foundation, Seattle, Washington, United States of America
¤e Current address: Icosavax, Inc., Seattle, Washington, United States of America
¶ Membership of HVTN 107 Protocol Team is provided in the Acknowledgements.
* zoe@fredhutch.org

## Abstract

### Background

Adjuvants are widely used to enhance and/or direct vaccine-induced immune responses yet rarely evaluated head-to-head. Our trial directly compared immune responses elicited by

**Data Availability Statement:** Data and protocols for HVTN 107 will be made publicly available online at https://atlas.scharp.org/cpas/project/HVTN% 20Public%20Data/begin.view.

**Funding:** This work was supported by the National Institute of Allergy and Infectious Diseases (NIAID) U.S. Public Health Service Grants AI068614 [LOC: HIV Vaccine Trials Network], AI068635 [SDMC: HIV Vaccine Trials Network], AI068618 [LC: HIV Vaccine Trials Network; AI069501 and NIH P30 AI064518 [Duke CFAR], UM1 AI069453 [Soweto-Bara Clinical Research Site] Funding was provided to Novartis Vaccines and Diagnostics (now part of GSK) by NIH (HHSN272201600012C) for the production process development of two gp120 envelope proteins TV1.C and 1086.C, and by the Bill & Melinda Gates Foundation Global Health (Grant OPP1017604) for the manufacture and release of the gp120s clinical grade material. The content is solely the responsibility of the authors and does not necessarily represent the official views of the NIAID, the National Institutes of Health (NIH), or the Gates Foundation. The funders had no role in study design, data collection and analysis, or decision to publish. GlaxoSmithKline Biologicals SA was provided the opportunity to review a preliminary version of this manuscript for factual accuracy, but the authors are solely responsible for final content and interpretation.

**Competing interests:** MK and OVDM are employed by GSK and hold shares in the company. MK reports grants from Bill & Melinda Gates Foundation and grants from NIH during the conduct of the study. NKT was an employee of GSK Biologicals and Novartis Vaccines & Diagnostics during the time of the trial. All other authors have nothing to declare.

**Abbreviations:** AE, adverse event; AESI, adverse events of special interest; BAMA, binding antibody multiplex assay; CCID, cell culture infectious dose; ICS, intracellular cytokine staining; MFI, mean fluorescence intensity; FDA, Food and Drug Administration; IND, Investigational New Drug; NHP, nonhuman primate; PFS, polyfunctionality score; PP, per-protocol; SAE, serious adverse event; SANCTR, South African National Clinical Trials Registry; STI, sexually transmitted infection; TM, transmembrane.

MF59 versus alum adjuvants in the RV144-like HIV vaccine regimen modified for the Southern African region. The RV144 trial of a recombinant canarypox vaccine vector expressing HIV *env* subtype B (ALVAC-HIV) prime followed by ALVAC-HIV plus a bivalent gp120 protein vaccine boost adjuvanted with alum is the only trial to have shown modest HIV vaccine efficacy. Data generated after RV144 suggested that use of MF59 adjuvant might allow lower protein doses to be used while maintaining robust immune responses. We evaluated safety and immunogenicity of an HIV recombinant canarypox vaccine vector expressing HIV *env* subtype C (ALVAC-HIV) prime followed by ALVAC-HIV plus a bivalent gp120 protein vaccine boost (gp120) adjuvanted with alum (ALVAC-HIV+gp120/alum) or MF59 (ALVAC-HIV+gp120/MF59) or unadjuvanted (ALVAC-HIV+gp120/no-adjuvant) and a regimen where ALVAC-HIV+gp120 adjuvanted with MF59 was used for the prime and boost (ALVAC-HIV+gp120/MF59 coadministration).

## Methods and findings

Between June 19, 2017 and June 14, 2018, 132 healthy adults without HIV in South Africa, Zimbabwe, and Mozambique were randomized to receive intramuscularly: (1) 2 priming doses of ALVAC-HIV (months 0 and 1) followed by 3 booster doses of ALVAC-HIV+gp120/ MF59 (months 3, 6, and 12), $n = 36$; (2) 2 priming doses of ALVAC-HIV (months 0 and 1) followed by 3 booster doses of ALVAC-HIV+gp120/alum (months 3, 6, and 12), $n = 36$; (3) 4 doses of ALVAC-HIV+gp120/MF59 coadministered (months 0, 1, 6, and 12), $n = 36$; or (4) 2 priming doses of ALVAC-HIV (months 0 and 1) followed by 3 booster doses of ALVAC-HIV +gp120/no adjuvant (months 3, 6, and 12), $n = 24$. Primary outcomes were safety and occurrence and mean fluorescence intensity (MFI) of vaccine-induced gp120-specific IgG and IgA binding antibodies at month 6.5.

All vaccinations were safe and well-tolerated; increased alanine aminotransferase was the most frequent related adverse event, occurring in 2 (1.5%) participants (1 severe, 1 mild). At month 6.5, vaccine-specific gp120 IgG binding antibodies were detected in 100% of vaccinees for all 4 vaccine groups. No significant differences were seen in the occurrence and net MFI of vaccine-specific IgA responses between the ALVAC-HIV+gp120/MF59-prime-boost and ALVAC-HIV+gp120/alum-prime-boost groups or between the ALVAC-HIV +gp120/MF59-prime-boost and ALVAC-HIV+gp120/MF59 coadministration groups. Limitations were the relatively small sample size per group and lack of evaluation of higher gp120 doses.

## Conclusions

Although MF59 was expected to enhance immune responses, alum induced similar responses to MF59, suggesting that the choice between these adjuvants may not be critical for the ALVAC+gp120 regimen.

## Trial registration

HVTN 107 was registered with the South African National Clinical Trials Registry (DOH-27-0715-4894) and ClinicalTrials.gov (NCT03284710).

Author summary

### Why was this study done?

- Vaccines may use an adjuvant to help the body produce a stronger immune response.

- Results from animal studies suggested that the MF59 adjuvant generates better immunogenicity than the alum adjuvant when given as part of an HIV vaccine and could also allow a lower dose of protein to be used.

- Our clinical trial was done to directly assess in humans whether MF59 leads to better immune responses than alum when given with protein in a subtype C canarypox vaccine (ALVAC-HIV) prime followed by ALVAC-HIV plus a bivalent gp120 protein vaccine boost (gp120).

### What did the researchers do and find?

- Vaccines were safe and well-tolerated over the 18 months of follow-up.

- 100% of vaccinees had vaccine-specific gp120 IgG binding antibodies at month 6.5.

- Immune responses for the ALVAC-HIV+gp120/MF59 group and the ALVAC-HIV+gp120/alum group were similar.

### What do these findings mean?

- Contrary to expectation, the choice between MF59 and alum does not seem critical to the immune responses assessed in the peripheral blood for this subtype C ALVAC-HIV+gp120 prime-boost regimen.

- The main limitations of our study were the small vaccine group sample sizes and that higher doses of gp120 protein were not evaluated.

## Introduction

Of the 8 HIV-1 vaccine candidates studied in efficacy trials [1–8], only the RV144 regimen showed a significant reduction in HIV-1 acquisition with 60% (95% CI: 22, 80) estimated vaccine efficacy at month 12 [9], waning to 31.2% (95% CI: 1.1, 52.1) by month 42. The RV144 vaccine regimen consisted of replication-defective canarypox-HIV recombinant ALVAC-HIV vector (vCP1521) at months 0 and 1 followed by 2 doses of vCP1521 plus alum-adjuvanted AIDSVAX subtypes B/E HIV envelope (env) glycoprotein (gp120) at months 3 and 6. Following the RV144 efficacy announcement, the Pox Protein Public–Private Partnership (P5) [10] was formed to develop a vaccine regimen to improve upon RV144 and tailor it to the most common global HIV subtype: subtype C [10]. The resultant regimen of the replication-defective canarypox vaccine (ALVAC) plus recombinant glycoprotein 120 (gp120) protein incorporated regionally adapted HIV-1 subtype C sub-Saharan African strains [4], replaced the alum

adjuvant with MF59, and added a month 12 boost. Although the subtype C vaccine regimen had an acceptable safety profile and met the prespecified immunogenicity criteria in the HVTN 100 phase 1-2a trial [11], it showed no vaccine efficacy in the phase 2b/3 HVTN 702 trial in South African adults [4].

Differences between the vaccine efficacy and immune correlates of HIV-1 acquisition risk reported in the RV144 trial in Thailand and the HVTN 702 trial in South Africa of similar ALVAC/gp120 pox-protein HIV-1 vaccine regimens have left open questions for the HIV vaccine field [4,8]. The significant reduction in HIV-1 acquisition seen in the modified intent-to-treat analysis of RV144 was not observed in the subtype C adapted regimen in South Africa, possibly because of differences in the vaccine regimens (e.g., inserts, adjuvant, booster schedule), populations (e.g., HIV incidence, HIV exposure, host genetics), and/or circulating viruses.

In response to findings from nonhuman primate (NHP) studies, the phase 1-2a HVTN 107 trial was designed to directly compare antibody and cellular immune responses of the alum versus MF59 adjuvants for the subtype C ALVAC/gp120 pox-protein regimen in a southern African population [12]. NHP studies found that replacing alum in the RV144 vaccine regimen with the MF59 adjuvant enhanced the magnitude of both systemic and mucosal immune responses to HIV-1 envelope (Env). However, use of the original alum adjuvant conferred protection from low dose $SIV_{mac251}$ or SHIV(AD8) challenge, whereas MF59 did not confer protection [13–16]. Prior to the emergence of these data, plans were already in place to use MF59 as the adjuvant in the HVTN 702 efficacy trial based on meeting the prespecified immunogenicity criteria in HVTN 100 and the enhanced antibody responses and improved durability observed in influenza vaccines adjuvanted with MF59 [9]. Thus, the MF59 versus alum adjuvant question was addressed in the HVTN 107 trial, which began 9 months after HVTN 702 and ended 2 months before the early unblinding of HVTN 702 due to non-efficacy. HVTN 107 also sought to evaluate the safety and immunogenicity of concurrent administration of ALVAC-HIV and subtype C recombinant gp120 proteins with MF59 adjuvant delivered at months 0, 1, 6, and 12 as a way to further enhance immune responses [17–20]. Subsequent analyses of blood samples from participants in the RV144 and HVTN 702 trials highlighted the important role of IgG binding antibodies specific for Env variable regions 1 and 2 (V1V2) and vaccine-specific CD4+ T-cell responses in protection from HIV-1 acquisition [21,22]. HIV-1–specific IgG and IgG3 serum bAb responses to HIV-1 envelope antigens correlated with reduced HIV-1 acquisition in the RV144 trial [21–23], thus were of interest in our study. Here, we report here on the safety and immunogenicity of the HVTN 107 vaccine regimen in view of the data on correlates of risk. The delay in reporting these findings is due in large part to the COVID-19 pandemic; protocol team members were heavily engaged in COVID-19 vaccine efficacy trials [24] when the HVTN 107 database lock occurred on May 11, 2020.

## Methods

### Participants

HVTN 107 was a phase 1-2a partially double-blinded, randomized clinical trial conducted at community research sites in South Africa (Cape Town, eThekwini, Soweto, and Tembisa), Mozambique (Maputo), and Zimbabwe (Harare).

Volunteers were eligible for enrollment if they were healthy, aged 18 to 40 years, could give written informed consent, were not living with HIV, were deemed at low vulnerability for HIV acquisition, and had not previously received an HIV vaccine. Participants of child-bearing potential were required to be on contraception, not pregnant, and non-lactating.

Enrollment was monitored to ensure no more than 60% of trial participants of either sex at birth were enrolled.

Participants were followed for 18 months after the initial vaccination. Safety evaluations included physical examinations and standard clinical chemistry and hematological tests, urinalysis, as well as pregnancy tests for participants of child-bearing potential. Local and systemic reactogenicity were assessed for 3 days following each vaccination or until resolution. Adverse events (AEs) were reported over 30 days after each vaccination visit, with a subset of AEs being reported for the duration of the study (including serious AEs [SAEs], AEs of special interest [AESIs], new chronic conditions requiring medical intervention for ≥30 days, sexually transmitted infections [STIs], and AEs leading to early participant withdrawal or early discontinuation of study product administration). All safety data were monitored by the HVTN Safety Monitoring Board.

## Ethics statement

Initial and ongoing approvals of the study protocol and research review were provided by the Medicine Control Council (20141022), the Fred Hutchinson Cancer Research Center institutional review board (IRB), and local research ethics committees for each site: National Health Bioethics Committee (CNBS) (Ref 125/CNBS/2014), University of Cape Town Human Research Ethics Committee (Ref 790/2014), University of Witwatersrand Human Research Ethics Committee (Medical) (#141108), University of Kwazulu-Natal Biomedical Research Ethics Committee (Ref BFC453/14), University of Zimbabwe Joint Research Ethics Committee (Ref 186/15). All participants gave written informed consent in a language of their choice prior to implementation of study procedures.

HVTN 107 was prospectively registered with the South African National Clinical Trials Registry (SANCTR) on October 7, 2014. The SANCTR registry date of October 14, 2019 is incorrect as the system would not have allowed an anticipated start date and last follow-up date to have occurred prior to the date of registration. This error likely occurred during SANCTR's migration of all previously registered trials to a new platform. The protocol underwent 2 revisions prior to study start. Protocol version 3.0 was written on May 4, 2017, with approval received on June 13, 2017, before the first participant enrolled on June 19, 2017. The trial was also posted on the US registry, ClinicalTrials.gov on September 15, 2017. Prospective registration on ClinicalTrials.gov was not required given the trial was prospectively registered with SANCTR, was conducted entirely outside of the United States, and did not involve an Investigational New Drug (IND) submission to the US Food and Drug Administration (FDA).

## Randomization and masking

During the consent process, participants could choose whether to enroll in a study subset to provide additional samples for the assessment of innate and mucosal immune responses. Participants enrolled in this subset (*n* = 72) were randomized in a 1:1:1:1 ratio to intramuscular injection of: (1) 2 priming doses of ALVAC-HIV (months 0 and 1) followed by 3 booster doses of ALVAC-HIV+gp120/MF59 (months 3, 6, and 12); (2) 2 priming doses of ALVAC-HIV (months 0 and 1) followed by 3 booster doses of ALVAC-HIV+gp120/alum (months 3, 6, and 12); (3) 4 doses of ALVAC-HIV+gp120/MF59 coadministration (months 0, 1, 6, and 12); or (4) 2 priming doses of ALVAC-HIV (months 0 and 1) followed by 3 booster doses of ALVAC-HIV+gp120/no adjuvant (months 3, 6, and 12) (**Fig 1**). Participants not in the additional sample subset (*n* = 60) were randomized in a 3:3:3:1 ratio to the 4 vaccine groups as the adjuvanted groups were primary interest. The HVTN Statistical Data Monitoring Center produced the block-randomized sequences by computer-generated random numbers, provided to each

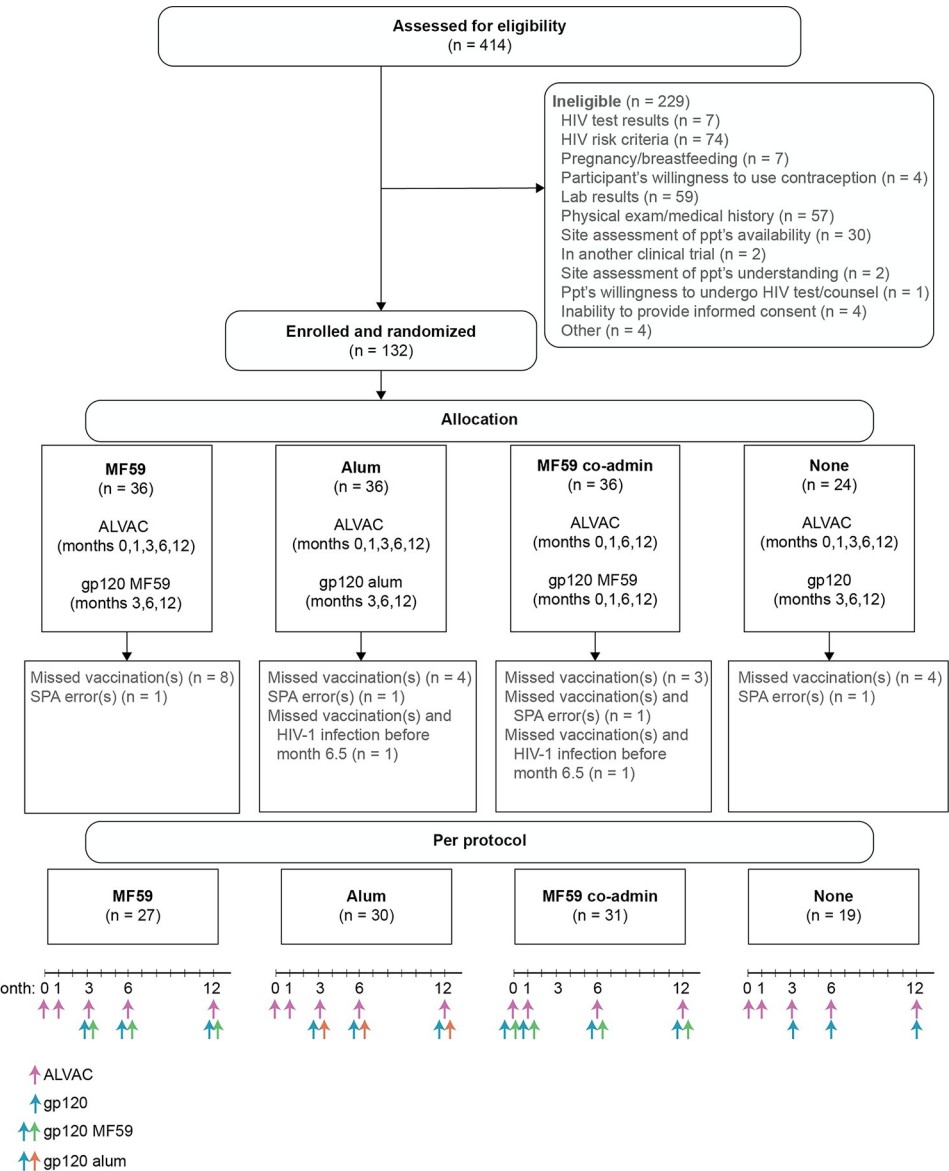

**Fig 1. CONSORT flow diagram: allocation and follow-up for HVTN 107.** HVTN, HIV Vaccine Trials Network; ppt, participant; SPA, study product administration.

study site through a web-based randomization system. Randomization was stratified by additional sample subset and country with blocks of size 4 (Mozambique, Zimbabwe) or 8 (South Africa) for the subset, or else blocks of size 10.

Participants, site staff (except for site pharmacists) who enrolled and followed participants, study team (except biostatisticians), and lab personnel were blinded to participant vaccine group assignments for the 3 prime-boost arms of the study with vaccinations at months 0, 1, 3, 6, and 12; the ALVAC-HIV+gp120/MF59 coadministration at months 0, 1, 6, and 12 was unblinded due to the different schedule. Site pharmacists were unblinded to ensure proper study product handling and dispensing, which included application of overlays to all syringes for site staff blinding.

## Study vaccines

The ALVAC-HIV (vCP2438) priming vaccine is an adaptation of the vCP1521 vector used in RV144 in which the CRF01_AE (subtype E) 92TH023 gp120 *env* insert was replaced with a subtype C gp120 *env* derived from strain 96ZM651, linked to the same transmembrane (TM) anchor sequence of *gp41* (derived from the subtype B strain LAI), and *gag* and *pro* derived from the subtype B strain LAI as in ALVAC-HIV vCP1521. ALVAC-HIV was administered at a dose of $10^7$ 50% cell culture infectious dose ($CCID_{50}$) and was donated by Sanofi Pasteur (Swiftwater, Pennsylvania).

The Env protein booster vaccine contains a bivalent subtype C gp120 Env derived from strains TV1.C and 1086.C, each at a dose of 100 μg. The gp120 protein vaccine was administered at a final dose of 0.5 ml with either MF59 oil-in-water emulsion or Aluminum Hydroxide Suspension (alum) adjuvant, containing 625 mcg of aluminum hydroxide. The MF59 adjuvant was provided by GSK Biologicals (Rixensart, Belgium), formerly Novartis (Cambridge, Maryland) and the alum adjuvant was provided by the Vaccine Research Center (NIAID, NIH), utilizing Rehydragel HPA from ChemTrade Logistics (Berkeley Heights, New Jersey, United States of America).

## Laboratory assays

All assays were performed blinded in HVTN laboratories utilizing validated methods [21,25–27]. Serum HIV-1–specific IgG binding antibody (bAb) responses were measured at 1:50 dilution, IgG3 at 1:40 dilution, and IgA at 1:10 dilution to vaccine-matched gp120 and V1V2 antigens by an HIV-1 binding antibody multiplex assay (BAMA) [21,23,28] (**S1 Text and S1 Table**). CD4+ T-cell responses to HIV vaccine insert-matched peptides were measured by intracellular cytokine staining (ICS) and analyzed by flow cytometry (**S1 Text and S1 Table**).

## Outcomes

The prespecified primary clinical objectives of the trial were to assess safety and tolerability of each vaccine regimen. Primary endpoints were local and systemic reactogenicity signs and symptoms, laboratory measures of safety, AEs, and SAEs. The prespecified primary immunogenicity objectives were to compare the IgG bAb response rates and magnitudes among positive responders to 1086, TV1, and ZM96 gp120 of the ALVAC-HIV+gp120/MF59 versus ALVAC-HIV+gp120/alum groups at month 6.5, and to compare IgA bAb response rates and magnitudes among positive responders to 1086, TV1, and ZM96 gp120 of the ALVAC-HIV+gp120/MF59 versus ALVAC-HIV+gp120/MF59 coadministration groups at month 6.5. Secondary objectives were to evaluate the IgG and IgG3 bAb responses to Env gp120 and V1V2 antigens and CD4+ T-cell responses to vaccine-matched antigens for each group at months 6.5, 12, 12.5, and 18.

## Statistical analysis

The safety evaluation included all enrolled participants and the immunogenicity analyses were performed on the per-protocol (PP) cohort, consisting of all participants who received the first 4 scheduled vaccinations, excluding participants who had an HIV-1 positive test prior to the immunogenicity blood draw.

The sample size of the trial provided at least 80% power to address the primary safety and immunogenicity objectives. For example, there is over 80% power to detect a 35% difference in the response rates between the groups primed with ALVAC followed by ALVAC + Bivalent

Subtype C gp120 and adjuvanted either with MF59 or with alum by a Fisher's exact two-sided test, with $n$ = 30 per group, assuming a 15% rate of missing data.

Immune responses were summarized by the proportion of participants with a positive response to individual antigens at each time point, with boxplots displaying the distributions of the immune response magnitudes among positive responders. Heatmaps were used to summarize the median magnitude for each antigen-specific CD4+ T-cell subset over vaccine groups and time points.

Kruskal–Wallis tests were used to compare the distributions of reactogenicity severity between vaccine regimens. Barnard's tests were used to compare the empirical immune response rates and Wilcoxon rank-sum tests to compare magnitudes among positive responders, COMPASS Env-specific CD4+ polyfunctionality scores [29], and area-under-the-magnitude-breadth curves (AUCMB) [30] for the comparisons of interest: ALVAC-HIV+gp120/MF59 versus ALVAC-HIV+gp120/alum groups (primary), ALVAC-HIV+gp120/MF59 versus ALVAC-HIV+gp120/MF59 coadministration groups (primary), ALVAC-HIV+gp120/MF59 versus ALVAC-HIV+gp120/no-adjuvant groups (secondary), and ALVAC-HIV+gp120/alum versus ALVAC-HIV+gp120/no-adjuvant groups (secondary).

The polyfunctionality score is defined as the estimated proportion of antigen-specific cell subsets detected, weighted by degree of functionality. The AUCMB is used as an aggregate measure of response against the panel of V1V2 antigens considered and is equivalent to the mean log bAb response across all antigens in the panel. Two-sided 95% CIs for positive response rates and differences in these were calculated using the score test method [31]. Two-sided 95% CIs for the difference in median magnitudes were calculated using the bootstrap percentile method with 1,000 bootstrap samples.

All $p$-values are two-sided, with $p$-values less than 0.05 deemed statistically significant. For each distinct hypothesis, the number of multiple tests was limited, and therefore multiplicity adjustments were not made. SAS (version 9.4; SAS Institute, Cary, North Carolina, USA) and R statistical software (version 4.0.4; R Foundation for Statistical Computing, Vienna, Austria) were used for statistical analysis. Further details can be found in S1 Text.

## Code

All statistical analyses were conducted using R (version 4.0.4, R Foundation for Statistical Computing, Vienna, Austria) and SAS (version 9.4, SAS Institute, Cary, North Carolina, USA) statistical software.

## Results

Between June 19, 2017 and June 14, 2018, a total of 132 participants were randomized and enrolled (80 from South Africa, 28 from Zimbabwe, and 24 from Mozambique). The ALVAC-HIV+gp120/MF59, ALVAC-HIV+gp120/alum, and ALVAC-HIV+gp120/MF59 coadministration groups consisted of 36 participants each and the ALVAC-HIV+gp120/no-adjuvant group consisted of 24 participants (Fig 1). Participants were followed for 18 months after the first vaccination with all participants completing the study by December 12, 2019. Sixty percent (79/132) of participants were female at birth, 100% were Black, and the median age was 25 years. Baseline characteristics and vaccination adherence were balanced across the treatment groups (Table 1). All 132 participants received their first vaccination, 81% (107/132) were in the PP cohort (received first 4 planned vaccinations), and 77% (101/132) received all scheduled vaccinations (Fig 1). HIV-1 was diagnosed in 5 participants on study; of these, 2 participants were diagnosed with HIV before month 6.5, 2 before month 12, and 1 participant before month 18, and were excluded from immunogenicity analyses at subsequent time points.

**Table 1. Baseline characteristics of the safety and PP cohorts.**

**A. Safety cohort, baseline characteristics and vaccination frequencies**

| | MF59 (n = 36) | Alum (n = 36) | MF59 Coadmin (n = 36) | None (n = 24) | Total (N = 132) |
|---|---|---|---|---|---|
| **Sex at birth** | | | | | |
| Male | 13 (36%) | 15 (42%) | 16 (44%) | 9 (38%) | 53 (40%) |
| Female | 23 (64%) | 21 (58%) | 20 (56%) | 15 (63%) | 79 (60%) |
| **Gender identity*** | | | | | |
| Male | 10 (28%) | 11 (31%) | 12 (33%) | 7 (29%) | 40 (30%) |
| Female | 18 (50%) | 17 (47%) | 16 (44%) | 13 (54%) | 64 (48%) |
| Transgender female | 0 (0%) | 0 (0%) | 0 (0%) | 0 (0%) | 0 (0%) |
| Transgender male | 0 (0%) | 0 (0%) | 0 (0%) | 0 (0%) | 0 (0%) |
| Gender variant | 0 (0%) | 0 (0%) | 0 (0%) | 0 (0%) | 0 (0%) |
| Self-identify | 0 (0%) | 0 (0%) | 0 (0%) | 0 (0%) | 0 (0%) |
| Prefer not to answer | 0 (0%) | 0 (0%) | 0 (0%) | 0 (0%) | 0 (0%) |
| Not assessed | 8 (22%) | 8 (22%) | 8 (22%) | 4 (17%) | 28 (21%) |
| **Age (years)** | | | | | |
| Median (range) | 24.0 (18–39) | 25.0 (18–37) | 25.0 (19–39) | 25.0 (18–33) | 25.0 (18–39) |
| **Vaccination frequency** | | | | | |
| Day 0 | 36 (100%) | 36 (100%) | 36 (100%) | 24 (100%) | 132 (100%) |
| Day 28 | 35 (97%) | 34 (94%) | 34 (94%) | 23 (96%) | 126 (95%) |
| Day 84 | 32 (89%) | 34 (94%) | - | 21 (88%) | 87 (91%)^ |
| Day 168 | 29 (81%) | 32 (89%) | 31 (86%) | 20 (83%) | 112 (85%) |
| Day 364 | 29 (81%) | 31 (86%) | 29 (81%) | 18 (75%) | 107 (81%) |

**B. PP cohort, baseline characteristics**

| | MF59 (n = 27) | Alum (n = 30) | MF59 Coadmin (n = 31) | None (n = 19) | Total (N = 107) |
|---|---|---|---|---|---|
| **Sex at birth** | | | | | |
| Male | 12 (44%) | 12 (40%) | 15 (48%) | 8 (42%) | 47 (44%) |
| Female | 15 (56%) | 18 (60%) | 16 (52%) | 11 (58%) | 60 (56%) |
| **Gender identity*** | | | | | |
| Male | 9 (33%) | 8 (27%) | 11 (35%) | 6 (32%) | 34 (32%) |
| Female | 10 (37%) | 14 (47%) | 12 (39%) | 9 (47%) | 45 (42%) |
| Transgender female | 0 (0%) | 0 (0%) | 0 (0%) | 0 (0%) | 0 (0%) |
| Transgender male | 0 (0%) | 0 (0%) | 0 (0%) | 0 (0%) | 0 (0%) |
| Gender variant | 0 (0%) | 0 (0%) | 0 (0%) | 0 (0%) | 0 (0%) |
| Self-identify | 0 (0%) | 0 (0%) | 0 (0%) | 0 (0%) | 0 (0%) |
| Prefer not to answer | 0 (0%) | 0 (0%) | 0 (0%) | 0 (0%) | 0 (0%) |
| Not assessed | 8 (30%) | 8 (27%) | 8 (26%) | 4 (21%) | 28 (26%) |
| **Age (years)** | | | | | |
| Median (range) | 26.0 (19–39) | 25.0 (18–37) | 25.0 (19–39) | 26.0 (18–33) | 25.0 (18–39) |

*Participants are not required to answer these questions, so counts may not match number of participants.

^Day 84% calculated based on 96 expected participants.

PP, per-protocol.

## Safety and reactogenicity

The vaccinations in all groups were well-tolerated with mild to moderate maximum systemic symptoms in 33.3% (44/132, 95% CI = 25.9%, 41.8%) of participants, mild to moderate injection site pain and/or tenderness in 48.5% (64/132, 95% CI = 40.1%, 56.9%), and mild to moderate injection site induration and/or erythema in 8.3% (11/132, 95% CI = 4.7%, 14.3%).

Maximum systemic symptoms refer to the maximum of the individual systemic symptoms (malaise and/or fatigue, myalgia, headache, nausea, vomiting, chills, and arthralgia) for a participant over all vaccinations. There were no significant differences in the distribution of reactogenicity severity observed among vaccine groups apart from temperature ($p$-value = 0.02), with higher temperatures seen in the ALVAC-HIV+gp120/alum group (**S1 Fig**).

AEs were reported by 73.5% (97/132, 95% CI = 65.4%, 80.3%) of participants. Three participants experienced AEs attributed to the study product by the investigator: 1 participant had increased alanine aminotransferase (severe); 1 participant had increased alanine aminotransferase (mild), increased aspartate aminotransferase (mild) and dermatitis (mild); 1 participant had rash (moderate) and rash maculo-papular (moderate); all resolved by the end of the trial. There were no related expedited AEs, AESIs, or pregnancies reported.

## Immunogenicity

### Primary outcomes

There were no significant differences in the IgG bAb response rates to vaccine-matched gp120 antigens between the ALVAC-HIV+gp120/MF59 and ALVAC-HIV+gp120/alum groups at month 6.5 (**Fig 2**). Median magnitudes were beyond the limit of linearity and could not be compared. Month 6.5 serum samples were titrated at serial dilutions of 1:200, 1:400, 1:800, and 1:1,600 to calculate geometric mean titer for the gp120 TV1 or 1086 proteins. No significant differences in geometric mean titer AUC were observed to either antigen at month 6.5 between the ALVAC-HIV+gp120/MF59 and ALVAC-HIV+gp120/alum groups (**S2 Fig**). No significant differences were seen in the IgA bAb response rates or magnitudes to vaccine-matched gp120 antigens between the ALVAC-HIV+gp120/MF59 and ALVAC-HIV+gp120/MF59 coadministration groups at month 6.5 (**Fig 3**).

### Secondary outcomes

Binding antibody responses were assessed to 5 envelope antigens from the following HIV-1 strains: ZM96 (ALVAC vaccine insert), 1086 and TV1 (protein inserts), A244 and B.CaseA V1V2 (protein boost inserts in the RV144 Thai trial that correlated with HIV-1 acquisition). Overall, gp120-directed responses to the vaccine-matched antigens were greater than the V1V2-directed responses, with considerable waning of all responses 6 months following the month 12 vaccination (**Figs 2, 4, S3, and S7**).

The use of either MF59 or alum adjuvant in the prime-boost regimen significantly increased the response rates at month 6.5 for IgG and IgG3 to the V1V2 variable region of HIV-1 envelope compared to the ALVAC-HIV+gp120/no-adjuvant group, with some higher response rates also induced by the ALVAC-HIV+gp120/alum group at months 12.5 and 18 also (**Figs 4 and S3**). Higher IgG response rates were observed for the ALVAC-HIV+gp120/alum group versus the ALVAC-HIV+gp120/no-adjuvant group to the 1086 V1V2 (difference = 36%, 95% CI = 6%, 58%, $p$-value = 0.022) and TV1 V1V2 (difference = 38%, 95% CI = 6%, 62%, $p$-value = 0.023) protein inserts at month 6.5 (**Fig 4**) as well as CaseA V1V2 (difference = 35%, 95% CI = 4%, 59%, $p$-value = 0.028, **S4 Fig**), with higher month 12.5 response rates to TV1 V1V2 (difference = 43%, 95% CI = 17%, 67%, $p$-value <0.001) and CaseA V1V2 (difference = 43%, 95% CI = 11%, 67%, $p$-value = 0.009). In addition, higher month 6.5 and 12.5 magnitudes among positive responders were seen to A244 V1V2 (median difference = 3878, 95% CI = 955, 5027, $p$-value = 0.002 and 2,311, 95% CI = 627, 8,158, $p$-value = 0.027, **S4 Fig**). The ALVAC-HIV+gp120/MF59 group also showed superior IgG response rates for 1086 V1V2 (difference = 44%, 95% CI = 14%, 65%, $p$-value = 0.006) and TV1 V1V2 (difference = 45%, 95% CI = 13%, 68%, $p$-value = 0.007, **Fig 4**) at month 6.5 relative

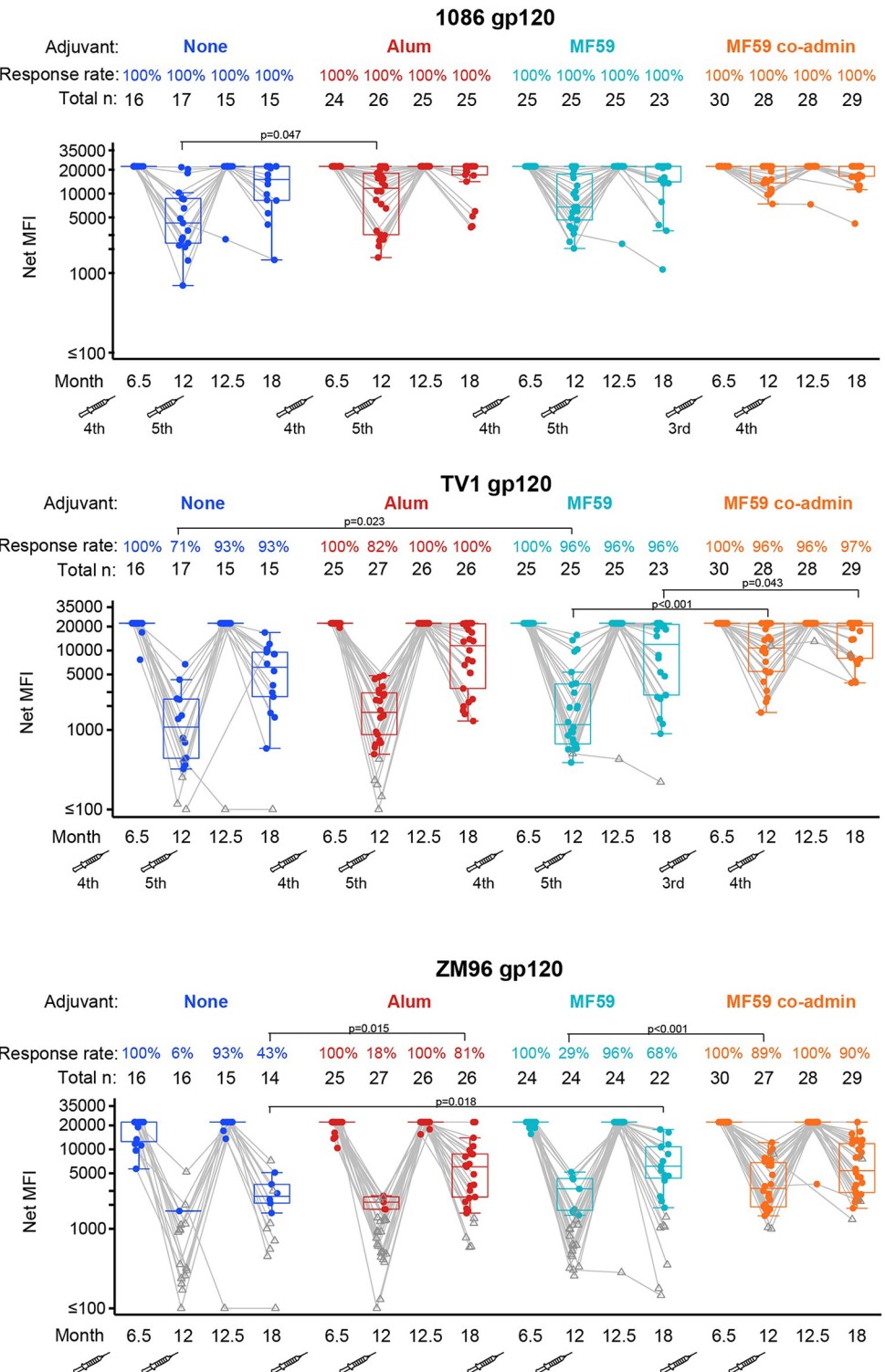

**Fig 2. Response rates and boxplots of binding antibody IgG responses to the 3 vaccine-matched gp120 antigens at months 6.5, 12, 12.5, and 18.** Significant *p*-values are shown for comparisons of response rates by Barnard's exact test and magnitudes among positive responders by Wilcoxon test for the following groups: MF59 vs. none, alum vs. none, MF59 vs. alum, MF59 vs. MF59 co-admin. The fourth and fifth vaccinations are shown by syringe. MFI, mean fluorescence intensity.

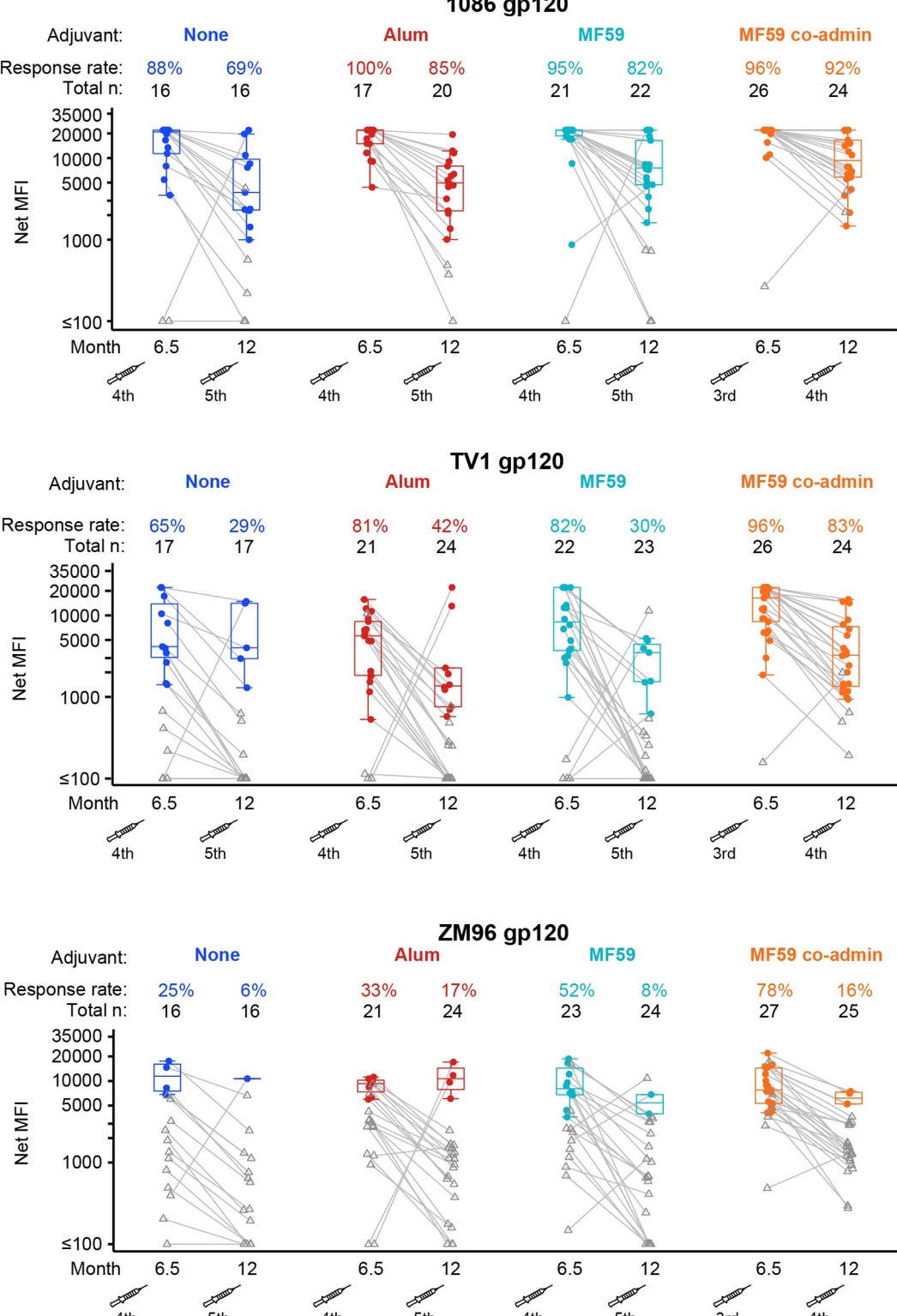

**Fig 3. Response rates and boxplots of binding antibody IgA responses to the 3 vaccine-matched gp120 antigens at months 6.5 and 12.** The fourth and fifth vaccinations are shown by syringe. MFI, mean fluorescence intensity.

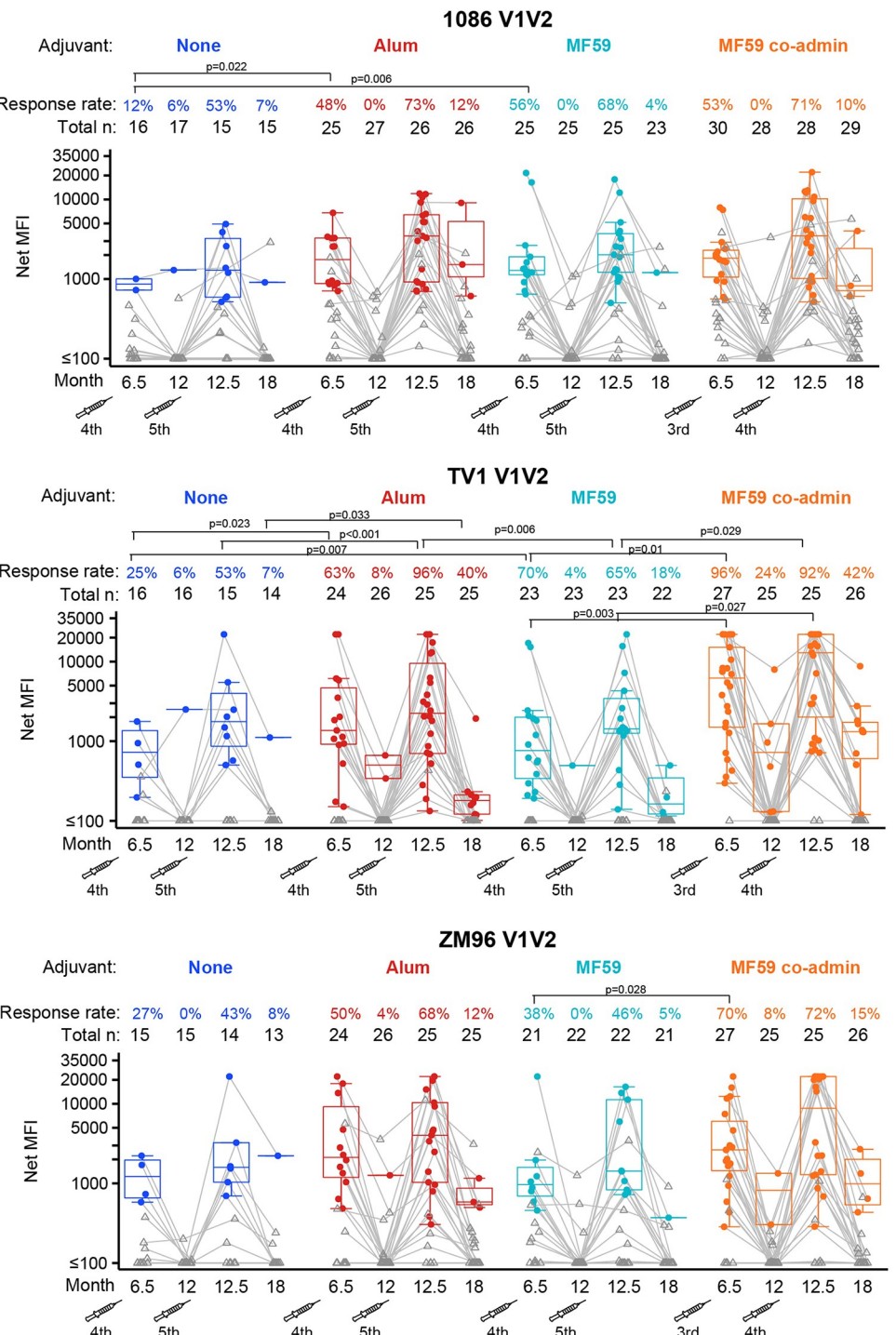

**Fig 4. Response rates and boxplots of binding antibody IgG responses to the 3 vaccine-matched V1V2 antigens at months 6.5, 12, 12.5, and 18.** Significant *p*-values are shown for comparisons of response rates by Barnard's exact test and magnitudes among positive responders by Wilcoxon test for the following groups: MF59 vs. none, alum vs. none, MF59 vs. alum, MF59 vs. MF59 co-admin. The fourth and fifth vaccinations are shown by syringe. MFI, mean fluorescence intensity.

to the ALVAC-HIV+gp120/no-adjuvant group as well as higher magnitudes among positive responders to A244 V1V2 at month 6.5 (median difference = 6743, 95% CI = 365, 8,320, *p*-value = 0.003, **S4 Fig**). IgG3 response rates to 1086 V1V2 at month 6.5 were significantly higher in both the ALVAC-HIV+gp120/alum and the ALVAC-HIV+gp120/MF59 groups than in the ALVAC-HIV+gp120/no-adjuvant group (difference = 40%, 95% CI = 12%, 61%, *p*-value = 0.008 and 28%, 95% CI = 2%, 50%, *p*-value = 0.036, respectively) (**S3 Fig**); IgG3 response rates to A244 V1V2 at month 6.5 were also significantly higher in the ALVAC-HIV+gp120/alum group than the ALVAC-HIV+gp120/no-adjuvant group (difference = 40%, 95% CI = 9%, 62%, *p*-value = 0.012). Looking more broadly at responses across the panel of 6 V1V2 antigens considered, higher IgG AUCMB for V1V2 antigens were seen for the ALVAC-HIV+gp120/MF59 group compared to the ALVAC-HIV+gp120/no-adjuvant group at month 6.5 and between the ALVAC-HIV+gp120/alum group compared to the ALVAC-HIV+gp120/no-adjuvant group at months 6.5 and 12.5 (**S5 Fig**) with no differences were seen for IgG3 (**S6 Fig**).

Response rates at durability time points (months 12 and 18, 6 months post-fourth and fifth vaccinations) were similarly improved by adjuvant, with a higher IgG rate for ALVAC-HIV+-gp120/alum versus ALVAC-HIV+gp120/no-adjuvant groups at month 18 to TV1 V1V2 (difference = 33%, 95% CI = 4%, 55%, *p*-value = 0.033, **Fig 4**) and ZM96 gp120 (difference = 38%, 95% CI = 7%, 64%, *p*-value = 0.015, **Fig 2**). A superior IgG response rate to TV1 gp120 was seen for the ALVAC-HIV+gp120/MF59 compared to the ALVAC-HIV+gp120/no-adjuvant group (difference = 25%, 95% CI = 4%, 50%, *p*-value = 0.023). No differences were seen in IgG or IgG3 AUCMB between the adjuvanted versus ALVAC-HIV+gp120/no-adjuvant groups at months 12 or 18 apart from higher IgG responses at month 18 for the ALVAC-HIV+gp120/alum group compared to the ALVAC-HIV+gp120/no-adjuvant group (**S5** and **S6 Figs**).

We next compared the 2 adjuvanted prime-boost groups, MF59 versus alum, and saw little difference in IgG and IgG3 bAb responses over time to various gp120 and V1V2 antigens apart from the following differences at month 12.5. The ALVAC-HIV+gp120/alum group had higher IgG response rates at month 12.5 to TV1 V1V2, CaseA V1V2, and A244 V1V2 (**Figs 4 and S4**). The ALVAC-HIV+gp120/alum group also had higher IgG and IgG3 AUCMB at month 12.5 (**S5 and S6 Figs**). IgG and IgG3 bAb response rates to the 3 gp120 vaccine-matched antigens were near 100% for both ALVAC-HIV+gp120/MF59 and ALVAC-HIV+-gp120/alum groups at months 6.5 and 12.5, with no significant difference at any time (**Figs 2 and S7**).

Next, we compared the coadministration of protein with MF59 to the MF59-prime-boost regimen. We found that coadministration generally resulted in significantly higher IgG response rates than seen in the ALVAC-HIV+gp120/MF59 group to the ZM96 V1V2 ALVAC vector insert, the TV1 V1V2 protein insert, and the CaseA V1V2 antigen at months 6.5 and 12.5, with no differences seen at month 18 (**Figs 4 and S4**). IgG TV1 V1V2 response rates were significantly higher in the ALVAC-HIV+gp120/MF59 coadministration group than the ALVAC-HIV+gp120/MF59 group at months 6.5 and 12.5 (difference = 27%, 95% CI = 7%, 48%, *p*-value = 0.010 and 27%, 95% CI = 4%, 49%, *p*-value = 0.029, respectively), with significantly higher magnitudes as well (median difference = 5441, 95% CI = 1,268, 8,014, *p*-value = 0.003 and 11,496, 95% CI = 663, 16,696, *p*-value = 0.027, respectively) (**Fig 4**). The ALVAC-HIV+gp120/MF59 coadministration group induced significantly higher response rates than the ALVAC-HIV+gp120/MF59 group to ZM96 V1V2 at month 6.5 (difference = 32%, 95% CI = 4%, 56%, *p*-value = 0.028) and to CaseA V1V2 at month 6.5 (difference = 58%, 95% CI = 33%, 76%, *p*-value <0.001) and month 12.5 (difference = 36%, 95% CI = 10%, 59%, *p*-value = 0.008), with significantly higher magnitudes at month 12.5 (median difference = 8,484, 95% CI = 661, 17,504, *p*-value = 0.013) (**Figs 4 and S4**). The ALVAC-HIV+-gp120/MF59 coadministration group also had higher IgG and IgG3 AUCMB at months 6.5

and 12.5 than the ALVAC-HIV+gp120/MF59 group as well as higher IgG AUCMB at month 18 (**S5** and **S6 Figs**). The IgG response rate to ZM96 gp120 was higher in the ALVAC-HIV+-gp120/MF59 coadministration group than in the ALVAC-HIV+gp120/MF59 group at month 12 (difference = 60%, 95% CI = 34%, 77%, $p$-value <0.001) with significantly higher magnitudes to TV1 gp120 at month 12 (median difference = 9,600, 95% CI = 6,550, 17,585, $p$-value <0.001), with some evidence at month 18 also (median difference = 8,611, 95% CI = −9,403 to 17,259, $p$ = 0.043) (**Fig 2**). No significant differences were observed between the geometric mean titer AUC to either TV1 gp120 or 1086 gp120 in the ALVAC-HIV+gp120/MF59-adjuvanted and ALVAC-HIV+gp120/MF59 coadministration groups from month 6.5 titrated serum samples (**S2 Fig**). IgG3 responses were also similar between the 2 groups (**S3 and S7 Figs**).

The use of either adjuvant in the context of the ALVAC+gp120 protein prime-boost regimen increased response rates of CD4+ T cells expressing IFN-γ and/or IL-2 and/or CD40L (the combination that detects most responding cells) to peptide pools covering the Env gp120 1086 and TV1 proteins but not to the gp120 ZM96 vector insert (**Fig 5**). No differences were observed in the CD4+ T-cell response magnitudes among positive responders to each vaccine-matched peptide pool between groups at any time point (**Fig 5**). CD4+ T-cell polyfunctionality scores (PFS) to the protein boosts, estimated by COMPASS [29] across 5 functions (IFN-γ, IL-2, TNF, CD40L, and IL-17), were generally increased by the use of the alum adjuvant, whereas MF59 only increased PFS to the TV1 gp120 protein (**S8 Fig**). CD4+ T-cell response rates to Gag were low, ranging from 0% to 11.8% (2/17) across all groups and time points (**S2 Table**), reflecting responses to the ALVAC vaccine alone and the importance of the protein boost, which included Env. CD8+ T-cell responses were also infrequent, with the highest response rates seen to TV1 gp120 where all response rates were at or below 12.5% (2/16) (**S3 Table**).

Higher response rates were seen in the ALVAC-HIV+gp120/alum group to 1086 and TV1 gp120 compared to the ALVAC-HIV+gp120/no-adjuvant group over time (**Fig 5**); PFS for the ALVAC-HIV+gp120/alum group to 1086 gp120 were also higher compared to the ALVAC-HIV+gp120/no-adjuvant group at months 12 and 12.5 and to TV1 gp120 at months 12, 12.5, and 18 (**S8 Fig**). The ALVAC-HIV+gp120/MF59 group had higher PFS than the ALVAC-HIV+gp120/no-adjuvant group for TV1 gp120 at months 12.5 and 18 (**S8 Fig**) and also induced superior CD4+ T-cell response rates to 1086 gp120 at month 12 compared to the ALVAC-HIV+gp120/no-adjuvant group (difference = 32%, 95% CI = 3%, 42%, $p$-value = 0.036, **Fig 5**), but not at other time points or to other antigens.

There were no significant differences in PFS between the ALVAC-HIV+gp120/MF59 and the ALVAC-HIV+gp120/alum groups at any time point (**S8 Fig**). These groups also had similar CD4+ T-cell responses at all time points, apart from months 12 and 18 (**Fig 5**). At months 12 and 18, the ALVAC-HIV+gp120/alum group had a higher response rate to TV1 gp120 than the ALVAC-HIV+gp120/MF59 group (month 12 difference = 32%, 95% CI = 5%, 54%, $p$-value = 0.019; month 18 difference = 30%, 95% CI = 2%, 53%, $p$-value = 0.039) and a higher response rate to 1086 gp120 at month 18 (difference = 30%, 95% CI = 3%, 54%, $p$-value = 0.032), with no other differences observed between these 2 groups. Differences were observed in the median magnitudes of individual cellular profiles: e.g., to 1086 gp120 at month 18 where IFN-γ+TNF+CD40L+ appears to be higher for the ALVAC-HIV+gp120/MF59 group and IL-2+TNF+CD40L+ higher for the ALVAC-HIV+gp120/alum group (**S9 Fig**).

Coadministration of protein resulted in a significant reduction in CD4+ T-cell response rates to the ZM96 gp140 peptide pool covering the vector insert at months 12.5 and 18 compared to the ALVAC-HIV+gp120/MF59 group (difference = −35%, 95% CI = −57%, −8%, $p$-value = 0.01 and −40%, 95% CI = −61%, −15%, $p$-value = 0.002, **Fig 5**). In addition, CD4+ PFS to ZM96 gp140 were lower for the ALVAC-HIV+gp120/MF59 coadministration group

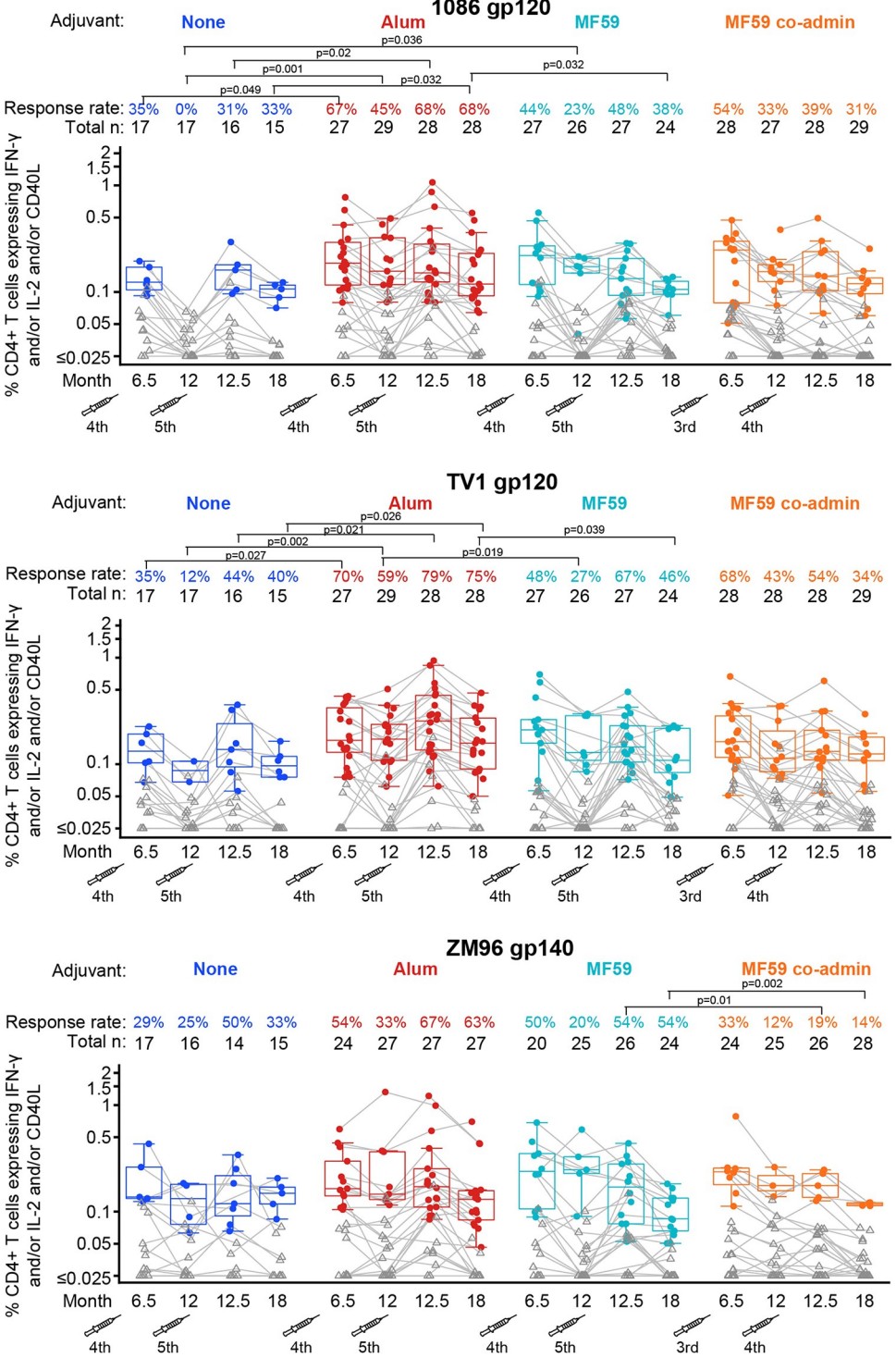

**Fig 5. Response rates and boxplots of CD4+ T-cell responses (measured by expression of IFN-γ and/or IL-2 and/or CD40L) to vaccine-matched antigens at months 6.5, 12, 12.5, and 18.** Significant *p*-values are shown for comparisons of response rates by Barnard's exact test and magnitudes among positive responders by Wilcoxon test for the following groups: MF59 vs. none, alum vs. none, MF59 vs. alum, MF59 vs. MF59 co-admin. The fourth and fifth vaccinations are shown by syringe.

compared to the ALVAC-HIV+gp120/MF59 group at all 4 time points, as well as at months 12.5 and 18 for 1086 and TV1 gp120 (S8 Fig).

Across all groups, polyfunctional CD4+ T cells expressing 2, 3, and 4 functional markers were detected. Highest magnitude cellular profiles included cells expressing 2 functions (CD40L with IL-2 or TNF), 3 functions (IL-2, TNF, and CD40L), and 4 functions that also includes IFN-γ. The magnitudes of CD4+ T cells expressing these 4 functions were visibly lower (S10 Fig) for the ALVAC-HIV+gp120/MF59 coadministration group compared to the ALVAC-HIV+gp120/MF59 and ALVAC-HIV+gp120/alum groups.

## Discussion

This study sought to determine whether vaccine-induced immune responses could be improved by 3 distinct factors: the use of adjuvant, the type of adjuvant, and the early coadministration of the gp120 protein with the recombinant canarypox ALVAC vaccine. Particular focus was given to the impact of these factors on the main correlates of HIV-1 acquisition risk identified in the RV144 Thai trial [21], the only prior preventative HIV vaccine clinical trial to demonstrate efficacy against infection.

A primary aim of our study was to compare immune responses induced by the alum and MF59 adjuvants in the ALVAC-HIV+gp120 prime-boost vaccine regimen, which could help elucidate whether the choice of adjuvant might explain the disparate vaccine efficacy results in the RV144 Thai trial and the HVTN 702 trial conducted in South Africa [4,8]. NHP studies of the 2 adjuvants showed reduced risk of SIV$_{mac251}$ acquisition for the ALVAC-HIV+gp120/alum vaccine compared to no protection for ALVAC-HIV+gp120/MF59 regimen, despite higher systemic immunogenicity of the latter, which also demonstrated higher mucosal antibodies targeting the V2 loop [13]. Other studies have shown that MF59 and alum influence different immunological spaces via CXCL10 and IL-1β, with MF59 associated with decreased risk of tier 1 SHIV-C acquisition [13,32]. Unlike the NHP studies' findings [13,32,33], our study detected some early immunological differences favoring the alum- rather than the MF59-adjuvanted group, with higher antibody response rates for certain V1V2 antigens as well as higher CD4+ T-cell response rates to the TV1 antigen. Such findings, however, were inconsistent across HIV antigens and time points and thus seem unlikely to play a significant role in the efficacy differences observed between the 2 ALVAC + gp120 protein regimen vaccine efficacy trials, RV144 and HVTN 702. Nonetheless, NHP data on an alum-prime-boost ALVAC regimen have shown that Env-specific CD4+T-cells play a critical role in containing virus replication at the mucosal level for the prevention of SIV$_{mac251}$ acquisition [34]. Given the finding that circulating IgA significantly correlated with decreased vaccine efficacy [21,28] and that IgA also negatively modified infection risk by antibody Fc effector functions [35], we evaluated whether there was a differential induction of IgA by MF59 and alum. In this study, there was no significant difference in IgA magnitude. Another study directly compared different adjuvants with the same immunogen and found that an adjuvant alone could not favorably alter the balance of IgG/IgG3 with IgA [36].

Another key question arising from our study is how the immune responses of the ALVAC-HIV+gp120/MF59 and ALVAC-HIV+gp120/alum groups compare to what was observed in the HVTN 100 gatekeeper trial of the ALVAC-HIV+gp120/MF59 vaccine regimen that preceded the HVTN 702 efficacy trial. Interestingly, only the ALVAC-HIV+gp120/MF59 group would have met the criteria prespecified in HVTN 100 to proceed to efficacy testing in HVTN 702 [11]. While both the adjuvanted groups in our study would have met the criteria for IgG vaccine-matched gp120 response rates at month 6.5, only the ALVAC-HIV+gp120/MF59 group would have met it for IgG V1V2 responses with a 70% response rate to TV1 V1V2. This

is similar to the response rate of the MF59-adjuvanted DNA prime-boost regimens studied in HVTN 111 (73%) although HVTN 111 response rates to 1086 V1V2 and B.CaseA V1V2 were notably higher than those observed with our ALVAC-HIV+gp120/MF59 group: 96% versus 56% and 60% versus 35% [37]. CD4+ T-cell response rates for the needle delivery MF59-adjuvanted DNA prime-boost arm of HVTN 111 [37] were comparable to those of the ALVAC-HIV+gp120/MF59 regimen in our study.

Our study also addressed the question of prime-boosting versus coadministration of ALVAC with the MF59-adjuvanted protein. Coadministration enhanced IgG antibody response rates to the V1V2 protein for all HIV antigens that were tested. As expected, such findings were most pronounced at the earlier time points and differences tended to diminish by month 18, 6 months following the final vaccination. Coadministration generally did not induce significantly higher IgG3 responses but negatively impacted CD4+ T-cell responses most notably to the ZM96 ALVAC insert, possibly due to antigenic competition [38–41]. We have observed lower CD4+ T-cell responses with coadministration compared to prime-boost regimens in 2 other published HVTN trials [42,43].

As adjuvants are known to increase local and systemic side effects, it is somewhat surprising that reactogenicity was remarkably similar between the adjuvanted and unadjuvanted groups. These data suggest that any reactogenicity observed is primarily driven by responses to the ALVAC vector, which itself is known to trigger innate immune signaling pathways [44]. Differences were however noted in the frequency and magnitude of bAbs directed against gp120 antigens, V1V2-directed antibodies as well as CD4+ T-cell responses, which were lower in the unadjuvanted group, highlighting the importance of comprehensive immune evaluations when testing adjuvants in clinical studies.

The current study has a number of limitations that temper conclusions especially as they apply to the findings of the prior efficacy studies of these vaccine regimens. First, the relatively small number of participants per group implies small but potentially scientifically relevant differences might have been missed. Furthermore, the total protein dose used in the current study was 200 mcg (100 mcg per protein) compared to the 600 mcg used in the RV144 trial, which could in turn influence adjuvant effects. However, data from the phase 1/2 RV132 [45] and RV135 [45] trials suggest that MF59 allows the use of a lower protein dose than the alum adjuvant. The RV132 trial studied a similar regimen to RV144 but with a total protein dose of 200 mcg and the MF59 adjuvant in place of alum. Neutralizing antibody geometric mean titers were over 3 times higher than those of the 600 mcg total protein dose, ALVAC-HIV+gp120/alum regimen studied in RV135, the same regimen as utilized in RV144. Our study was limited to the primary and secondary outcomes of antibody binding and CD4+ T-cell responses. Further work might examine antibody Fc effector functions that can differ among adjuvants and the breadth of antibody recognition to the V2 loop of circulating strains [36,46]. Finally, we did not analyze mucosal responses that could also be differentially altered by adjuvant compared to immune responses measured in the blood.

Our findings suggest that the choice of alum versus MF59 adjuvant did not play a major role in modifying the magnitude of the IgG V1V2 and CD4+ T-cell responses. Further work is needed to understand the observed differential vaccine efficacy in the RV144 and HVTN 702 efficacy trials. It remains possible that the difference in adjuvant combined with differences in the vaccine constructs, host populations, and/or circulating HIV virus subtypes may have eliminated the partially protective vaccine efficacy seen in RV144.

## Supporting information

**S1 CONSORT Checklist. CONSORT 2010 checklist of information to include when reporting a randomized trial.**
(DOC)

**S1 Text. Supplemental methods.**
(PDF)

**S1 Table. Details of the BAMA and ICS antigens including HIV-1 viral strain information.**
(PDF)

**S2 Table. CD4+ T-cell responses (measured by expression of IFN-γ and/or IL-2 and/or CD40L) to Gag at months 6.5, 12, 12.5, and 18.**
(PDF)

**S3 Table. CD8+ T-cell responses (measured by expression of IFN-γ and/or IL-2 and/or CD40L) to TV1 gp120 at months 6.5, 12, 12.5, and 18.**
(PDF)

**S1 Fig. Frequency of maximum severity of systemic (A) and local (B) reactogenicity symptoms by group in the safety cohort.**
(PDF)

**S2 Fig. Serum IgG binding antibody geometric mean titers (AUC) of all participants in all groups at month 6.5.**
(PDF)

**S3 Fig. Response rates and boxplots of binding antibody IgG3 responses to the 3 vaccine-matched V1V2 and A244 V1V2 antigens at months 6.5, 12, 12.5, and18.**
(PDF)

**S4 Fig. Response rates and boxplots of binding antibody IgG responses to B.CaseA V1V2 and A244 V1V2 antigens at months 6.5, 12, 12.5, and 18.**
(PDF)

**S5 Fig. Comparison of serum V1V2 IgG binding antibody magnitude-breadth of geometric mean MFI (AUC).**
(PDF)

**S6 Fig. Comparison of serum V1V2 IgG3 binding antibody magnitude-breadth of geometric mean MFI (AUC).**
(PDF)

**S7 Fig. Response rates and boxplots of binding antibody IgG3 responses to the 3 gp120 vaccine-matched antigens at months 6.5, 12, 12.5, and 18.**
(PDF)

**S8 Fig. Distribution of CD4+ polyfunctionality score (PFS) to vaccine-matched antigens at months 6.5, 12, 12.5, and 18.**
(PDF)

**S9 Fig. Boxplots of magnitude of CD4+ T-cell response by marker subset to vaccine-matched antigens at months 18 for the marker subsets analyzed by COMPASS.**
(PDF)

**S10 Fig. Heatmaps of median magnitude of CD4+ T-cell response by marker subset to vaccine-matched antigens at months 6.5, 12, 12.5, and 18 for the marker subsets analyzed by COMPASS.**
(PDF)

## Acknowledgments

We thank the study participants, colleagues, and staff on the protocol team for their support as well as all Community Advisory Board members who contributed to this study. We thank Yong Lin, Lu Zhang, Dr. Nicole Yates, Sheetal Sawant, David Beaumont, Mark Sampson, Angelina Sharak, Judith Lucas, Michael Archibald (BAMA team), Dr. Marcella Sarzotti-Kelsoe (Duke QAU) and Dr. Kevin Saunders, Dr. Barton Haynes (Duke Human Vaccine Institute), and the Protein Production Facility at Duke University for protein reagent design and production; Asiphe Besethi, Mahlodi Montlha, Boitumelo Mosito, Shamiska Rohith, Saleha Omarjee, Stephany Wilcox (ICS team), Sarah Everett, Bronwill Herringer, Chadwin Rushin (CHIL operations), and Nicolette Schuller (CHIL QA); Daryl Morris (Fred Hutchinson Cancer Center). The HVTN 107 Protocol Team: Paul Goepfert, University of Alabama, Birmingham, Birmingham, AL, USA; Kathryn Mngadi, Aurum Institute, South Africa; Londiwe Luthuli and Diantha Pillay, CAPRISA, South Africa; Nicole Grunenberg, Zoe Moodie, Erica Andersen-Nissen, Simbarashe Takuva, On Ho, Laurie Rinn, Jill Zeller, Xue Han, Gina Escamilla, Carter Bentley, Ingrid Durrenberger, Nandi Luthuli, Michelle Nebergall, and Erik Schwab, Fred Hutchinson Cancer Center, Seattle, WA, USA; Laura Polakowski, Irene Rwakazina, and Michael Pensiero, DAIDS, NIAID, Bethesda, MD, USA; Thoko Norah Sifunda, PHRU, South Africa; Lindiwe Mbhele, UKZN, South Africa; Sanjay Phogat and Carlos Diazgranados, Sanofi Pasteur, Swiftwater, PA, USA; Marguerite Koutsoukos and Olivier Van Der Meeren, GSK Vaccines, Belgium.

## Author Contributions

**Conceptualization:** Nicole Grunenberg, Carlos A. Diazgranados, Marguerite Koutsoukos, Olivier Van Der Meeren, Susan W. Barnett, Niranjan Kanesa-thasan, Georgia D. Tomaras, M. Juliana McElrath, Lawrence Corey.

**Data curation:** Zoe Moodie, Erica Andersen-Nissen, Jia J. Kee, Maurine D. Miner.

**Formal analysis:** Zoe Moodie, Erica Andersen-Nissen, Jia J. Kee.

**Funding acquisition:** Georgia D. Tomaras, M. Juliana McElrath, Lawrence Corey.

**Investigation:** Nicole Grunenberg, One B. Dintwe, Faatima Laher Omar, Linda-Gail Bekker, Fatima Laher, Nivashnee Naicker, Ilesh Jani, Nyaradzo M. Mgodi, Portia Hunidzarira, Modulakgota Sebe, Laura Polakowski, Shelly Ramirez, Michelle Nebergall, Simbarashe Takuva, Lerato Sikhosana, Jack Heptinstall, Kelly E. Seaton, Stephen De Rosa, Kathryn Mngadi, Paul Goepfert.

**Methodology:** Zoe Moodie, Erica Andersen-Nissen, Georgia D. Tomaras.

**Project administration:** Shelly Ramirez, Michelle Nebergall, Simbarashe Takuva.

**Software:** Zoe Moodie, Jia J. Kee.

**Writing – original draft:** Zoe Moodie, Erica Andersen-Nissen, Kathryn Mngadi, Paul Goepfert.

**Writing – review & editing:** Zoe Moodie, Erica Andersen-Nissen, Nicole Grunenberg, One B. Dintwe, Faatima Laher Omar, Jia J. Kee, Linda-Gail Bekker, Fatima Laher, Nivashnee Naicker, Ilesh Jani, Nyaradzo M. Mgodi, Portia Hunidzarira, Modulakgota Sebe, Maurine D. Miner, Laura Polakowski, Shelly Ramirez, Michelle Nebergall, Simbarashe Takuva, Lerato Sikhosana, Jack Heptinstall, Kelly E. Seaton, Stephen De Rosa, Carlos A. Diazgranados, Marguerite Koutsoukos, Olivier Van Der Meeren, Susan W. Barnett, Niranjan Kanesa-

thasan, James G. Kublin, Georgia D. Tomaras, M. Juliana McElrath, Lawrence Corey, Kathryn Mngadi, Paul Goepfert.

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
