## [Editor Report · Decision Letter 0]

13 Dec 2023

Dear Dr Moodie, 

Thank you for submitting your manuscript entitled "Safety and immune responses among healthy adults without HIV in HVTN 107: A phase 1/2a randomized trial of a subtype C ALVAC-HIV (vCP2438) and bivalent subtype C gp120 vaccine adjuvanted with MF59 or alum" for consideration by PLOS Medicine.

Your manuscript has now been evaluated by the PLOS Medicine editorial staff and I am writing to let you know that we would like to send your submission out for external peer review.

Please re-submit your manuscript within two working days, i.e. by Dec 15 2023 11:59PM.

Kind regards,

Katrien G. Janin, PhD

Senior Editor

PLOS Medicine

---

## [Decision Letter · Decision Letter 1]

31 Jan 2024

Dear Dr. Moodie,

Thank you very much for re-submitting your manuscript "Safety and immune responses among healthy adults without HIV in HVTN 107: A phase 1/2a randomized trial of a subtype C ALVAC-HIV (vCP2438) and bivalent subtype C gp120 vaccine adjuvanted with MF59 or alum" (PMEDICINE-D-23-03685R1) for review by PLOS Medicine.

I have discussed the paper with my colleagues and the academic editor and it was also seen again by the reviewers. I am pleased to say that provided the remaining editorial and production issues are dealt with we are planning to accept the paper for publication in the journal.

[LINK]

We expect to receive your revised manuscript within 8 days. Please email us (plosmedicine@plos.org) if you have any questions or concerns, or like to request an extension.

If you have any questions in the meantime, please contact me (kjanin@plos.org) or the journal staff on plosmedicine@plos.org.  

We look forward to receiving the revised manuscript by Feb 12 2024 11:59PM.   

Sincerely,

Katrien Janin, PhD

Senior Editor 

PLOS Medicine

plosmedicine.org

Comments from Reviewers:

Reviewer #1: Statistical review

This is a paper reporting a phase I/II trial comparing four different formulations of HIV-1 vaccine in terms of safety and vaccine response. I had reviewed the previous submission reporting this trial. Generally I found this resubmission easier to follow and my previous comments were addressed well.. The issues with registration of the trial raised also appear to be addressed.

I have a few additional minor comments on this submission.

1. Introduction, line 155 "waning to 31.2% (95% CI: 1.1, 52.1, p-value = 0.04)" - I would remove the p-value here (or add the p-value to the previous line's result) to keep consistent.

2. Methods line 247: it would be useful to provide some intuition as to why the ratio was 3:3:3:1 for individuals not in the additional sample subset.

3. Line 328-329: personally I would recommend the sample size calculation is featured in the main paper.

4. Line 475 - I think there's a typo here with the median difference and 95% CI for month 18.

5. Line 495: I do not think PLOS medicine permits 'data not shown'.

James Wason

Reviewer #2: Dear authors and editor:

All raised comments have been addressed and implemented properly in the revised version of the manuscript. Results of the study are now better presented, more structured and the manuscript has overall improved in its clarity for the reader. The work address a very critical question in the field for vaccine regimen design; both in terms of adjuvant selection and prime/boost timings. Although sample size is an important limitation of the trial, it is unlikely that a bigger trial would be ever be designed and would provide different evidence that the current trial. Overall I think it's worth its publication. Thanks for the opportunity to review this work. 

Reviewer #3: The MS is very much improved. How ever one minor point is that the antigenic contents in GP 120 C plus MF 59 is only 100 mcg. the gp 120 B/E plus Alum used in RV 144 was 300 mcg.

Not sure whether authors would like to add discussion on this.

Requests from Editors:

Editorial requests:

1. Title: We suggest that the title is modified slightly to better conform to PLOS Medicine style. Eg, “Safety and immunogenicity of a subtype C ALVAC-HIV vaccine prime plus bivalent subtype C gp120 vaccine boost adjuvanted with MF59 or alum in health adults without HIV (HVTN 107): A phase 1/2a 2 randomized trial” (or similar).

2. Abstract/background: Regarding this text, “Our trial directly compared immune responses elicited by alum vs. MF59 adjuvants in the RV144-like HIV vaccine regimen modified for the Southern African region; RV144 is the only trial to have shown modest HIV vaccine efficacy. Data generated after RV144 suggested that use of MF59 adjuvant might allow lower protein doses to be used.”

a. The composition of the RV144 vaccine needs to be introduced briefly in the abstract (as you have done in the introduction);

b. Please indicate whether the RV144 vaccine regimen was adjuvanted with alum, as is implied in the Introduction, lines 160-161 (“…replaced the alum adjuvant with MF59…”);

c. Please add an additional sentence to explain the change of adjuvant in the new regimen with regard to the expectation that MF59 would enhance immunogenicity and why (ie, results from NHP studies). This addition is also needed to put into context the first sentence of the Abstract conclusion (otherwise this is the first mention of the expectation that MF59 would enhance responses).

3. Abstract/background, lines 90-91: Please delete “with one less dose” for simplicity; this detail is made clear later and isn’t needed to understand the trial design;

4. Abstract, lines 102-103: Please slightly revise to “Primary outcomes were safety and occurrence and mean fluorescence intensity (MFI) of…”

5. Author summary/What did the researchers do and find: “100% of all vaccinees” should be revised to “All vaccines” or “100% of vaccines” (ie, “all” and “100%” are redundant).

6. Introduction, first sentence: Please briefly introduce the RV144 vaccine (vaccine components and regimen) for those who are not familiar with the details of these trials (eg, ALVAC-HIV at mo 0 and 1, ALVAC-HIV + bivalent gp120 boost at mo 3 and 6).

7. Introduction, line 165: I would slightly rephrase this sentence for clarity; eg, “ Differences between the vaccine efficacy and immune correlates of HIV-1 acquisition risk reported in the RV144 trial in Thailand and the HVTN 702 trial in South Africa (and other similar pox-protein HIV-1 vaccine regiments) have left open questions for the HIV vaccine field.”

8. Introduction, line 172: Please add references 13-16 at the end of this statement for clarity: “In response to findings from nonhuman primate (NHP) studies…”

9. Introduction, lines 177-178: “However, use of the original alum adjuvant conferred protection from low dose SIVmac251 or SHIV(AD8) challenge.” Did the RV144 regimen adjuvanted with MF59 *not* confer protection in NHP, or was this not tested?

10. Introduction, line 182: “…the MF59 vs. alum adjuvant question was addressed in the parallel HVTN 107 trial.” By ‘parallel’ do you mean parallel to HVTN 702? It would be useful if this was made more explicit that these two trials ran in parallel.

11. Introduction, lines 185-188: This sentence is confusing as written; I suggest dividing it into 2 sentences. Eg, “Subsequent analyses of blood samples from participants in the RV144 and HVTN 702 trials highlighted the important role of IgG binding antibodies specific for Env variable regions 1 and 2 (V1V2) and vaccine-specific CD4+ T-cell responses in protection from HIV-1 acquisition [21,22]. Here we report here on the safety and immunogenicity of the HVTN 107 vaccine regimen in view of those data on correlates of protection.”

12. Methods, line 239: Please clarify the following sentence: “During the consent process, participants could choose whether to enrol in a subset to provide additional samples.” I would suggest something like, “During the consent process, participants could choose whether to enrol in a study subset to provide additional samples for the assessment of innate and mucosal immune responses.”

13. Method (general): please add sample size calculation, demonstrating that the trial was adequately powered. Please move relevant text from supplement to main text.

14. Results, line 347. Please move Table S2 to the main body of the manuscript. The baseline characteristics table should be presented in the main paper, per CONSORT.

15. Results, overall primary and secondary outcomes sub-headers comment: I wonder if it may be better to have one single ‘Primary outcomes’ and one single ‘Secondary outcomes’ sub-header without the additional information being part of the sub-headers, as this feels not needed (and makes for heavy reading). Please remove all other sub-headers in the Immunogenicity section (apart from one single ‘Primary outcomes’ and one single ‘Secondary outcomes’). 

16. Results, lines 388-390: The following sentence should be moved to the introduction: “HIV-1–specific IgG and IgG3 serum bAb responses to HIV-1 envelope antigens correlated with reduced HIV-1 acquisition in the RV144 trial [21, 28, 32] and were therefore of interest in our study.” In addition, the wording makes it sound as if this were tested post-hoc, but these endpoints were pre-specified, so you might wish to rephrase this.

17. Results, line 393-395, please include a call-out to the appropriate figure referenced in this statement: “Overall, gp120-directed responses to these antigens were greater than the V1V2-directed responses, with considerable waning of all responses 6 months following the month 12 vaccination.”

18. Results, line 454. This sentence starts with “Finally, we compared …” yet in the next paragraph you compare T-cell responses in ALVAC-HIV+gp120/MF59 or ALVAC482 HIV+gp120/alum vs. ALVAC-HIV+gp120/no-adjuvant groups. Please amend or make clear by the first 3 comparatives are more closely related. 

19. Results, line 495. Data not shown is not permitted. Please add data to support this statement, or remove the statement. 

20. Discussion, lines 535-536: The following sentence should be removed from the discussion: “The publication of NHP studies that assessed the effect of alum and MF59 adjuvants on immunogenicity and SHIV/SIV acquisition motivated the HVTN 107 study”. This fits better in the introduction, where it is more elaborated upon. You can start this section with “ This study sought to determine … .” (line 536).

21. Discussion, line 553. “. Unlike the NHP studies’ findings, …” Please insert reference(s) for the NHP studies. 

22. Figures. There are a large number of figures currently included in the main paper. Please consider whether any of the figures could be moved to the supplement.

23. Please ensure you have redefined abbreviations in the captions of your figures and legends (e.g see S1 Table). 

24. Please remove the financial statement from the end of the manuscript and include this only in the relevant parts of the manuscript submission form. The financial statement will be compiled as metadata.

25. Please remove page numbers from the CONSORT checklist (your published article will not contain page numbers)

[LINK]

---

## [Editor Report · Decision Letter 2]

14 Feb 2024

Dear Dr Moodie, 

On behalf of my colleagues and the Academic Editor, I am pleased to inform you that we have agreed to publish your manuscript "Safety and immunogenicity of a subtype C ALVAC-HIV (vCP2438) vaccine prime plus bivalent subtype C gp120 vaccine boost adjuvanted with MF59 or alum in healthy adults without HIV (HVTN 107): A phase 1/2a randomized trial" (PMEDICINE-D-23-03685R2) in PLOS Medicine.

PRESS

Sincerely, 

Katrien G. Janin, PhD 

Senior Editor 

PLOS Medicine